# Investigation of the Resistance to High-Speed Impact Loads of a Heterogeneous Materials Reinforced with Silicon Carbide Fibers and Powder

**DOI:** 10.3390/ma16020783

**Published:** 2023-01-13

**Authors:** Alexander Malikov, Alexander Golyshev

**Affiliations:** Khristianovich Institute of Theoretical and Applied Mechanics SB RAS, 4/1 Institutskaya Str., Novosibirsk 630090, Russia

**Keywords:** ceramic fiber, SiC, high-speed impact, additive technologies, microstructure, cratering

## Abstract

Pioneering studies on the additive manufacturing of a cermet heterogeneous material using SiC ceramic fiber were carried out. Unique studies of the damage staging (cratering) and the transition to the destruction of the formed material during high-speed impact created with the help of an electrodynamic mass accelerator have been carried out. It has been shown that the use of ceramic fiber in a metal matrix reduces the impact crater depth by 22% compared to material with ceramic particles. For the first time, the phase composition of the resulting composite was studied using synchrotron radiation. It was shown that, as a result of laser exposure, silicon carbide SiC is dissolved in the titanium matrix with the formation of secondary compounds of the TiC and Ti5Si3C types. It has been established that the use of SiC ceramic fibers leads to their better dissolution, in contrast to the use of SiC ceramic particles, with the formation of secondary phase compounds, and to an increase in mechanical characteristics.

## 1. Introduction

Parts made of titanium alloys are widely used in the automotive, military, and aerospace industries due to their light weight, high hardness and strength, and corrosion resistance [1]. Refractory ceramic materials, such as titanium boride, silicon carbide, silicon nitride, and aluminum oxide, appear to be promising heat-resistant components of cermet materials for titanium matrices [2,3,4]. The development of composite materials began in the 1960s with well-known works [5,6] when experiments were carried out with metal-matrix composites (MMCs). The use of ceramic and carbon fibers as a filler is currently mainly used in the manufacture of composite materials by sintering, under conditions of high mechanical and thermal loads [7,8]. In [9], for the first time, composites of the carbon fiber-titanium matrix type were obtained using a new liquid-phase process using an intermediate matrix with a melting temperature lower than that of titanium. SiC fibers have excellent properties such as high strength, high modulus, excellent heat resistance and oxidation resistance, etc. [10,11]. Therefore, fibers have wide prospects for application in the field of aviation, astronautics, nuclear energy, and weaponry. The use of fibers in MMC should lead to an increase in tensile strength, impact strength, modulus of elasticity, and resistance to mechanical stress during high-speed impact.

The use of fibers should lead to an increase in tensile strength, impact strength, modulus of elasticity and resistance to mechanical stress in high-speed impact. Due to the fact that heterogeneous materials with fibers are formed mainly by sintering technology, the dimensions of the created products are determined by the dimensions of the molds, which, in turn, are small and wear out quickly. A well-suited solution to this problem is additive manufacturing technology, which involves rapid, localized heating using laser radiation. Research is underway to produce bulk cermets using additive manufacturing [12,13,14,15].

One of the widely used and most promising methods of additive manufacturing is direct metal deposition (DMD) technology. High efficiency is the main advantage of DMD; however, such problems as internal defects, low mechanical properties and control of the geometrical accuracy of the geometric dimensions of printed parts remain. Also, porosity defects (microvoids) are inevitable when printing with metal, which can be characteristically divided into pores as a result of non-melting, gas pores and keyhole pores [16]. One of the solutions is the use of ultrasonic vibrations. It was shown in [17] that the combination of ultrasonic and traditional laser cladding not only effectively prevents defects, but also improves the microstructure and increases the microhardness.

A small amount of work is devoted to the production of metal-matrix material based on titanium alloy and SiC ceramics in the form of particles and fibers by additive technologies.

In [18], a composite based on Ti2AlNb reinforced with SiC fiber was obtained by selective laser melting. This composite has increased microhardness, strength, and no cracks. In [19], to create a composite material consisting of a metal matrix reinforced with SiC fibers, a hybrid approach was used, combining laser cladding and hot isostatic pressing. Parts were formed from a fiber-reinforced titanium alloy with a fiber volume fraction of 17%. However, it should be noted that in [19] post-processing in the form of hot isostatic pressing was used, which complicates the overall manufacturing process. The work is devoted to the creation of composites based on a titanium alloy with SiC ceramic particles using selective laser melting technology [20]. An increase in microhardness and wear resistance was shown. However, comparative studies of the effect of SiC ceramic fibers or SiC ceramic particles on mechanical impact resistance in high-speed impact have not been investigated.

In this work, ceramic-metal heterogeneous materials were obtained by additive technologies using SiC fiber and SiC ceramic particles as reinforcing elements and Ti64 alloy.

## 2. Materials and Methods

Three types of materials were used in this work: Ti64 titanium alloy powder (Figure 1a), silicon carbide ceramic powder (Figure 1b), and silicon carbide ceramic fibers (Figure 1c). Metal-ceramic mixtures were subsequently formed from these materials and deposited on a substrate in the form of a plate of titanium alloy Ti-6.5Al-1.8Zr-1.5Mo with dimensions of 50 × 50 × 10 mm.

For creating a ceramic-metal heterogeneous material by laser surface cladding (LSC) and direct laser deposition (DMD), an IPG Photonics ytterbium fiber laser with a wavelength of 1.07 μm and a maximum power of 3 kW was used. Figure 2 is a schematic representation of the used laser technologies.

SiC-Ti64 powder mixture with a ratio of 1:9 wt was prepared by mechanical mixing in a Venus FTLMV-02 V-mixer for one hour until a homogeneous powder mixture was formed. The SiC–Ti64 powder cermet mixture was deposited on the substrate by the DMD method (laser power was 1000 W, scanning speed was 16 mm/s, laser spot diameter was approximately 7 mm, powder flow rate was 7 g/min, gas flow rate was 10 l/min) to form a coating with a thickness of approximately 2700 µm.

A fundamentally different method was used to deposit the ceramic-metal mixture with fibers. In this case, the LSC method was used. Moreover, at the beginning, ceramic fibers were ground in a high-energy planetary mill “Activator-2SL” for 40 s to a size of 300–600 microns in length, and then they were poured on the substrate with the first sublayer 150 microns thick. Next, a second sublayer of Ti64 powder 250 µm thick was poured above. To form a multilayer material, this cycle was repeated until a coating of approximately 2700 µm thick was formed (laser power was 1000 W, scanning speed was 16 mm/s, laser spot diameter was approximately 7 mm).

The microstructure was studied using a Zeiss EVO MA 15 scanning electron microscope in the backscattered electron (BSE) mode. Experimental studies of the impact resistance of the created coatings were carried out using the process of high-speed interaction between the impactor and the target. The impactors were accelerated using electrodynamic mass acceleration (EDMA). The principle of operation of EDMA is based on the occurrence of an electromagnetic force acting on a conductive object located between the conductive walls of the channel (rails) when current flows through the formed circuit. In this work, a plasma piston was used as a conductive body, which exerts pressure on the accelerated container. A steel ball weighing 0.5 g was used as a impactor. The interaction between the impactor and the ceramic-metal coating was carried out at speed of 1150 m/s. The speed of the body on the free flight path was determined by the time of passing markers located at a distance of 100 mm from each other. High-speed video filming of the flight process was carried out using a Photron SA-Z high-speed camera with a frequency of 100 kHz at a frame exposure time of 1 μs.

To determine the phase composition, X-ray phase analysis was carried out on a D8 Advance X-ray diffractometer using the characteristic radiation of the copper anode of the Cu-Kα X-ray tube (λ = 1.5406 Å), a nickel filter to suppress the reflection from Cu-Kβ radiation and a linear-position sensitive detector Lynx-Eye. The phase composition was deciphered using the PDF4 database.

To determine the phase compounds in the bulk of the material, a study was carried out using synchrotron radiation (SR) at VEPP-3 located at the Institute of Nuclear Physics G.I. Budker SB RAS at the “Hard X-Ray Diffractometry” station. A 1-mm thick sample was studied using SR with a wavelength of 0.3685 Å in the Debye-Cherard geometry. The survey was carried out sequentially, starting from the upper layer, in the range from 0° to 22° with a step of 200 μm. The beam diameter was 100 μm.

## 3. Results and Discussion

Figure 3 shows an electron microscope image of a metal matrix multilayer material with SiC ceramic particles and SiC ceramic fibers at 250× and 2000× magnification.

In the metal-matrix coating obtained from Ti64 + SiC particles (Figure 3a), a more uniform distribution of reinforcing elements is observed, compared with the sample with ceramic fibers (Figure 3b). In both cases, the size of SiC decreases under the exposure to laser radiation. According to the literature data, SiC ceramics reacts with a titanium alloy melt, which leads to the formation of secondary phases in situ [18,20].

Figure 4 shows x-ray diffraction patterns (XRD). The formation of secondary phase compounds of the Ti5Si3 intermetallic type and Ti0.8V0.2C0.6 carbide was found in the sample.

The table shows the results of quantitative analysis obtained using XRD patterns presented in Figure 4. According to the results, it was found that for a sample with a ceramic fiber, the compounds Ti5Si3 and Ti0.8V0.2C0.6 (Table 1), are formed more actively. As a result, it can be assumed that the fibers dissolve in the metal matrix better than the ceramic particles. However, with these processes, XRD data obtained with the reflection method may not provide a complete picture. It is important to understand what is happening in the volume of the test sample. To solve this problem a synchrotron radiation was used, which, due to its advantages, can examine through a sample up to 2 mm thick.

According to the images of diffraction circles obtained using synchrotron radiation, diffractograms were formed for Ti64-SiC and Ti64-crushed fiber SiC (Figure 5) The phase composition was deciphered using the PDF4 database.

Based on the results of interpretation of the diffraction patterns, it was shown that both for the ceramic powder and for ceramic fibers, silicon carbide SiC was dissolved in the titanium matrix with the formation of secondary compounds of the TiC and Ti5Si3Cx type, which were not originally presented. New phases were synthesized in a metal matrix as a result of a chemical reaction between elements (8Ti + 3SiC→Ti5Si3 + 3TiC) during laser exposure. This reaction is possible due to the negative Gibbs free energy, ΔG, which is the maximum energy that is available to do useful work as a result of a chemical reaction. However, when comparing the reflections of diffraction patterns for samples of Ti64 with SiC powder and Ti64 with crushed fiber SiC it can be seen that the fiber dissolves better than the particles (see Figure 5). When ceramic powder is used, the TiC reflections are characterized not only by their lower intensity, but also by their number compared to the sample of ceramic particles. The result obtained can be explained by the fact that the elongated shape of the fiber has a free surface that reacts with the titanium matrix. As a result, the ceramic fiber more actively enters into a chemical reaction with the formation of secondary phase compounds.

Using ceramic fiber induces more active filling the metal matrix with secondary-formed reinforcing compounds, which causes an increase in mechanical characteristics. Figure 6 presents photos of the studied samples with prints from the indenter (Figure 6a,c) and the dependence of the micro-hardness of the cermet coating on the removal from the substrate (Figure 6b,d). Measurements were carried out for a matrix with a step of 200 μm between prints.

For the Ti64 sample with crushed fiber SiC, the first half of the sample is characterized by a micro-hardness of 550 HV0.3. However, then there is a gradual increase in micro-hardness to a value of 943 HV0.3 due to secondary phase formations in this area (Figure 6b). When using ceramic powder a fundamentally different picture is observed. Figure 6 shows that at different points of the measurement sample, the microhardness varies from 354 HV0.3 to 713 HV0.3, and the average value is 512 HV0.3.

Due to the fact that the formed ceramic-metal composite materials are multilayer structures with significantly different morphology (see Figure 3), it is necessary to carefully study the mechanical characteristics on a higher scale. To assess the mechanical properties of a particular layer, as well as plotting the dependence of mechanical properties on the area of measurement, instrumental indentation of the deposited structure along the thickness was performed using a NanoScan nanohardness tester.

Figure 7 shows changes in hardness (Figure 7a) and modulus of elasticity (Figure 7b) for a material with SiC fibers. Indentation was carried out according to the method of Oliver-Farr and GOST. A series of 4 × 100 injections was performed with a step of 20 μm with a force of 0.1 N. The modulus of elasticity for the Ti64 substrate was 147 ± 4 GPa, and the Berkovich hardness was 4.75 ± 0.47 GPa. First deposited layer has a high content of secondary phases. Its properties differ sharply from those of the substrate. In the deposited layer with a high ceramic content, the mechanical properties change—for example, the hardness varies from 9 ± 0.6 GPa to 13.5 ± 1.5 GPa, and the modulus of elasticity from 195 ± 15 GPa to 240 ± 20 GPa. Next layer has small amount of ceramic, and the hardness values vary from 6 ± 0.4 GPa to 9 ± 0.7 GPa and the elastic modulus from 150 ± 15 GPa to 175 ± 15 GPa. Near the edge of the substrate, there is a layer with a high content of ceramics; however, in this area, the greatest spread in the values of mechanical properties is noted (see Figure 7). This behavior is primarily due to the inhomogeneity of the material structure: high values correspond to undissolved ceramic particles, and low values correspond to the titanium matrix.

As for the mechanical characteristics of a sample based on silicon carbide ceramic particles, a different picture is observed. In this case, the mechanical properties of the deposited layers do not depend on the distance from the substrate, but only on the distribution of silicon carbide particles. The hardness for the deposited layers was 7.5 ± 0.7 GPa (see Figure 8a), and the values of the elastic modulus were 160 ± 15 GPa (see Figure 8b). When the indenter hit the silicon carbide ceramic particle, the values of hardness and elastic modulus significantly exceeded the mechanical characteristics of the deposited layer: hardness 35 ± 10 GPa (see Figure 8a), modulus 350 ± 50 GPa (see Figure 8b).

Figure 9 shows photographs of coatings after impact testing. As a result of the interaction of the impactor with the targets, a craters with a different character of destruction were formed.

It can be seen that the high-speed interaction of the impactor with the cermet coating results in the destruction of the deposited layer along the crater perimeter (see Figure 9). For a sample with ceramic particles, the destruction occurred in the form of a hole with chipping (Figure 9a). For a sample with a ceramic fibers, destruction occurs in the form of disk formation (Figure 9b). The result obtained can be explained by the fact that the use of ceramic fiber led to an enhanced formation of secondary phases in contrast to the use of fiber and, as a result, to an increase in mechanical characteristics. The modulus of elasticity of ceramics E determines its rigidity, and hence the speed of elastic waves that dissipate the impact energy over the volume of the sample. In addition, the increase in stiffness allows the material to better withstand bending stresses that occur near the point of impact and lead to cracking. Therefore, an increase in the modulus of elasticity had a beneficial effect on the ballistic resistance of the sample. In this case, it should be taken into account that with increasing modulus, the acoustic impedance of the material also increases, which, in turn, affects the interaction of the ceramic-metal layer with the impactor and substrate.

This is consistent with the literature data. It is known that the ballistic response after impact tests depends on the structural-phase composition of the alloys [21]. Thus, targets with a lamellar microstructure exhibited brittle disk fracture. Single-phase alloys with equiaxed microstructure are destroyed due to the ductile growth of holes.

However, if we examine the crater depth, then for the sample with SiC powder the crater depth is 1437 µm. In the case of using ceramic fiber, the crater depth is 1137 µm, 22% less. The results obtained allow us to conclude that the coating with fibers ensures efficient dissipation of the kinetic energy of the impactor over the entire volume of the sample, and not only near the impact point.

## 4. Conclusions

For the first time, a ceramic-metal heterogeneous material was formed by additive manufacturing using a titanium alloy matrix reinforced with SiC ceramic fiber. Unique studies of the damage staging (cratering) and the transition to the destruction of the formed material during high-speed impact created with the help of an electrodynamic mass accelerator have been carried out.

For the first time, the phase composition of the obtained composite was studied using synchrotron radiation at a megascience facility (BINP SB RAS). It was shown that, as a result of laser exposure, silicon carbide SiC was dissolved in the titanium matrix with the formation of secondary compounds of the TiC and Ti5Si3C types. It has been established that the use of SiC ceramic fibers leads to their better dissolution, in contrast to the use of SiC ceramic particles, which leads to an increase in mechanical characteristics.

It is shown that the use of ceramic fiber in metal matrix allowed to decrease crater depth by 22% and compared to the sample with SiC particles.

A coating with ceramic fibers, in contrast to ceramic particles, provides a more efficient dissipation of the kinetic energy of the indenter throughout the entire volume of the sample, and not just near the point of impact.

## Figures and Tables

**Figure 1 materials-16-00783-f001:**
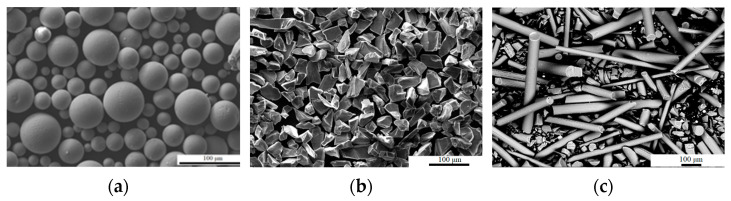
SEM image of the used powders ((**a**)-Ti64, (**b**)-SiC ceramic particles, (**c**)-SiC ceramic fibers).

**Figure 2 materials-16-00783-f002:**
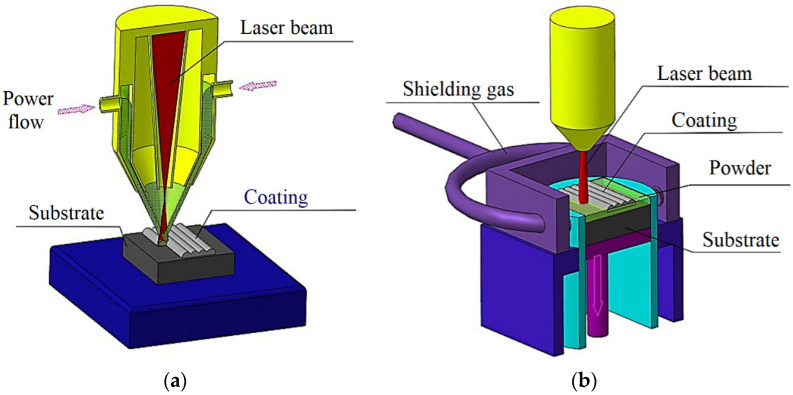
Schematic representation of laser technologies: (**a**)-direct metal deposition technology (DMD) (**b**)-laser surface cladding technology (LSC).

**Figure 3 materials-16-00783-f003:**
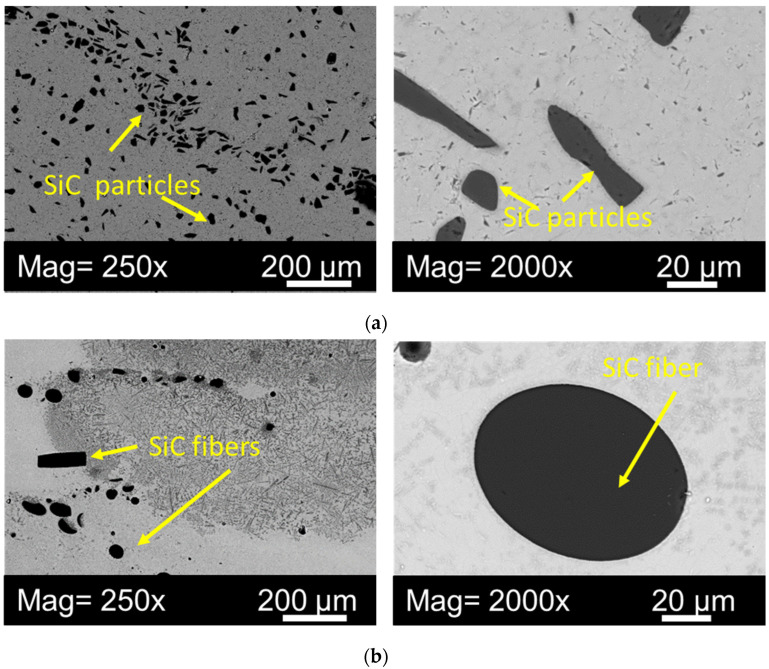
SEM image of a metal-matrix multilayer material ((**a**)-Ti64 + SiC ceramics (DMD), (**b**)-Ti64 + SiC ceramic fibers (LSC).

**Figure 4 materials-16-00783-f004:**
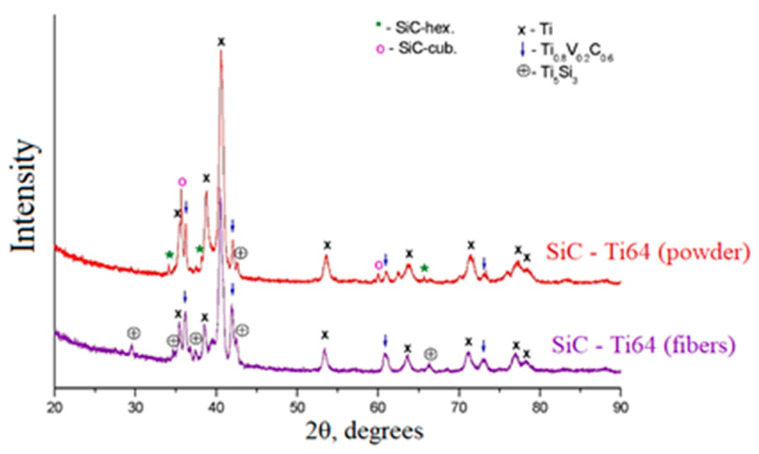
X-ray diffraction patterns of Ti64-SiC powder and Ti64-crushed fiber SiC samples.

**Figure 5 materials-16-00783-f005:**
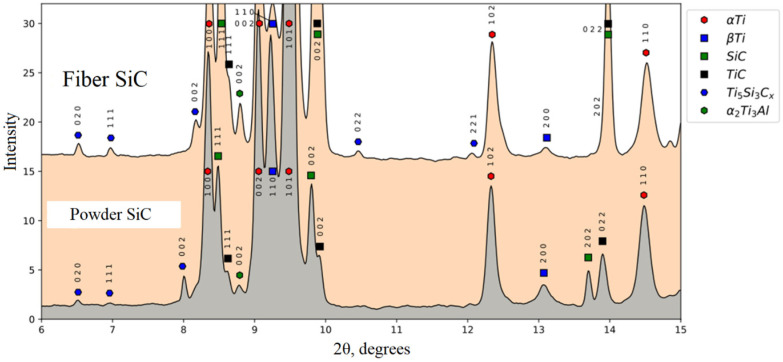
Diffraction patterns of the composite material Ti64-SiC powder and Ti64-crushed fiber SiC).

**Figure 6 materials-16-00783-f006:**
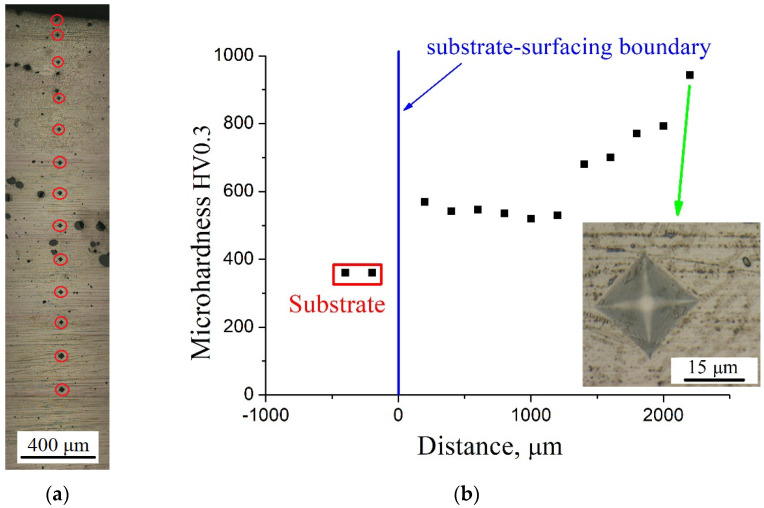
Photos of the indentor prints on the samples with the SiC fiber (**a**) and powder (**c**) and the dependence of the microhardness of the cermet coating on the graduate from the substrate for samples with the SiC fibers (**b**) and the SiC powder (**d**).

**Figure 7 materials-16-00783-f007:**
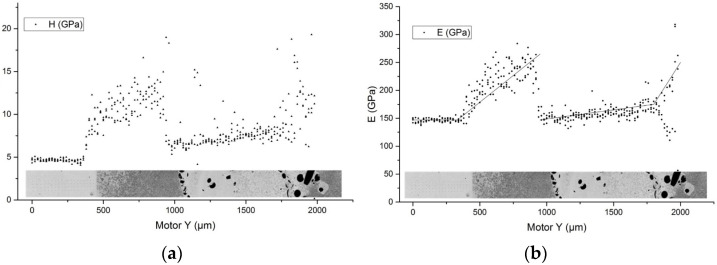
Dependence of hardness (**a**) and elastic modulus (**b**) on the place of measurement for the sample Ti64 + crushed fiber SiC.

**Figure 8 materials-16-00783-f008:**
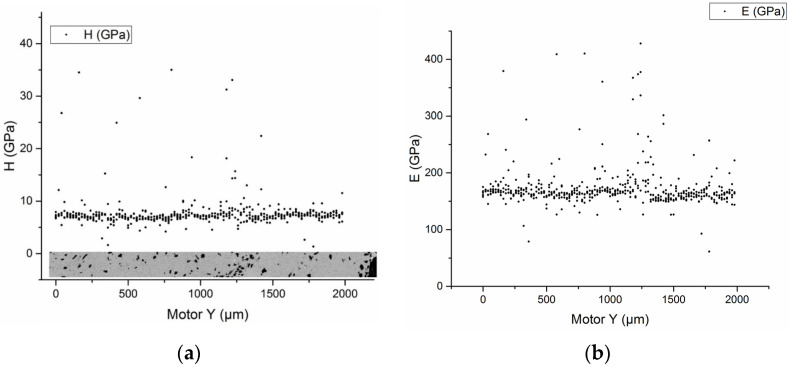
Dependence of hardness (**a**) and elastic modulus (**b**) on the place of measurement for the sample Ti64 + SiC powder.

**Figure 9 materials-16-00783-f009:**
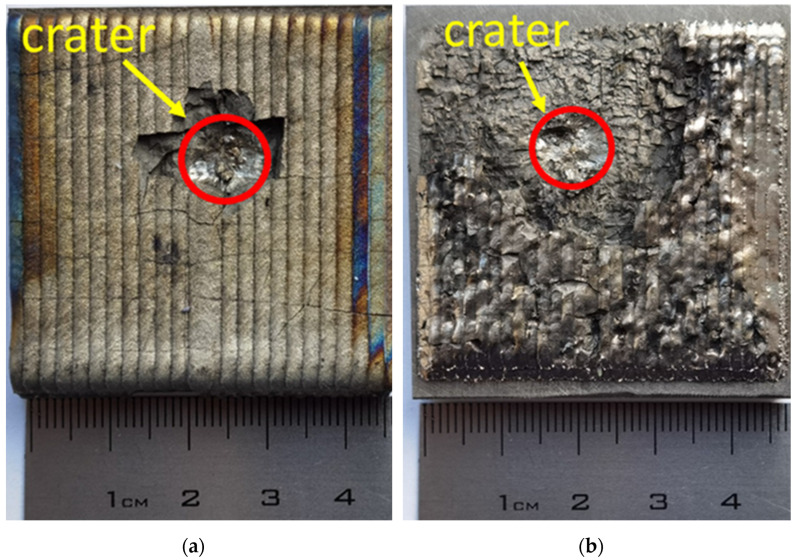
Photographs of the coating after impact testing: (**a**)-Ti64 + SiC ceramic particles (DMD), (**b**)-Ti64 + SiC ceramic fibers (LSC).

**Table 1 materials-16-00783-t001:** Quantitative results (wt.%).

Sample	SiC-Hex	SiC-Cub	Ti	Ti_0.8_V_0.2_C_0.6_	Ti_5_Si_3_	Note
Ti64–SiC (particles)	2	3	86	7	3	An unidentified phase is observed
Ti64–SiC (fibers)	–	–	67	20	13	An unidentified phase is observed

## Data Availability

Not applicable.

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
