# Peer review of "Investigation of the Resistance to High-Speed Impact Loads of a Heterogeneous Materials Reinforced with Silicon Carbide Fibers and Powder"

_materials, 2023, doi:10.3390/ma16020783_

Round 1

Reviewer 1 Report

The article is written properly and investigations were done appropriately

Few concerns 

 Literature regarding ceramic fibers needs to be added in the introduction part. Literature didn’t carry out well

Fig 1. The comparison scale should be of identical dimension , either all at 100micron

How the authors measured the impactor speed to 1150m/s exactly?

How much mass penalty was added once authors moved from particles to fibers?

How many samples had undergone impact testing?

It looks like only one, in such case it can be outliers, needs more data to conclude the results

However, if we examine the crater depth, then for the sample with SiC powder the 222

crater depth is 1437 μm. In the case of using ceramic fiber, the crater depth is 1137 μm, 223

22% less.

On the basis of a single experiment, how do the authors generalize 22% less?

Impact mechanism need to be discussed, why fibers are more dominating than particle, in terms of their mechanics and basic physics

Need more literature on impact characterization, particles, and fibers response to such impacts

Author Response

The authors are grateful to the reviewers for their time and consideration, as well as valuable suggestions for improving the manuscript. All changes in the text are highlighted in yellow.

Reviewer 1

Comments and Suggestions for Authors

The article is written properly and investigations were done appropriately

Few concerns 

Literature regarding ceramic fibers needs to be added in the introduction part. Literature didn’t carry out well

Response.

Agree with the comment. Literature on ceramic fibers has been added to the introduction.

Fig 1. The comparison scale should be of identical dimension, either all at 100micron

Response.

Agree with the comment. It was corrected.

How the authors measured the impactor speed to 1150m/s exactly?

Response.

The speed of the body on the free flight path was determined by the time of passing markers located at a distance of 100 mm from each other. High-speed video filming of the flight process was carried out using a Photron SA-Z high-speed camera with a frequency of 100 kHz at a frame exposure time of 1 μs. The relative error in measuring the distance between markers using a caliper is 0.1%. The error in determining the distance in the frame at a resolution of one pixel of about 0.3 mm is less than 1%, taking into account body blur during exposure. The accuracy of determining the time between frames is 0.0001%. Thus, the relative error in determining the body velocity in the measuring area is about 1%.

How much mass penalty was added once authors moved from particles to fibers?

Response.

The study of sections of samples showed that the area occupied by ceramic fibers in the cross section is 10% of the total area of surfacing. As a result, the calculated mass concentration of ceramic fibers is approximately 8 wt. %, however, this value is actually slightly higher because the fibers have an elongated shape. The concentration of SiC ceramic powder in the initial powder mixture is also 10 wt%.

How many samples had undergone impact testing? It looks like only one, in such case it can be outliers, needs more data to conclude the results “However, if we examine the crater depth, then for the sample with SiC powder the 222 crater depth is 1437 μm. In the case of using ceramic fiber, the crater depth is 1137 μm, 223 22% less.” On the basis of a single experiment, how do the authors generalize 22% less?

Response.

At the first stage of the work, a series of three shock tests of the created coating from the VT-6 titanium alloy was carried out. It was found that the measured values of the crater depths differ from each other by no more than 5%. The next stage of work was the addition of ceramics (powder and fibers) to the titanium alloy and the testing of the ceramic-metal coating. Based on the data obtained for the sample from VT-6, we assumed that for the ceramic-metal coating the scatter in the values of the crater depth would also be 5%. However, this fact, as well as the effect of ceramic concentration on cratering, requires a separate study and will be presented in the next paper.

Impact mechanism need to be discussed, why fibers are more dominating than particle, in terms of their mechanics and basic physics

Response.

The observed result can be explained by different structural-phase composition and, consequently, by different mechanical properties of the formed material. The modulus of elasticity of ceramics determines the rigidity of the material, and hence the speed of elastic waves that dissipate the impact energy over the volume of the sample. Increasing the stiffness allows the material to better withstand bending stresses that occur near the point of impact and lead to cracking. Therefore, an increase in the modulus of elasticity had a beneficial effect on the ballistic resistance of the sample. This discussion is included in the article.

Need more literature on impact characterization, particles, and fibers response to such impacts

Response.

The main direction of research in the creation of materials with ceramic particles is aimed at studying mechanical properties: strength and elastic characteristics, Young's modulus, hardness, crack resistance, which is shown, for example, in the following works:

https://www.sciencedirect.com/science/article/pii/S1005030219302403 https://www.sciencedirect.com/science/article/pii/S1359835X22000148?via%3Dihub

In this paper, we also explored some of these characteristics. It is important to note that high-speed impact tests for a material created using additive technologies have been done for the first time.

Reviewer 2 Report

Comments can be found in the attachment.

Author Response

The authors are grateful to the reviewers for their time and consideration, as well as valuable suggestions for improving the manuscript. All changes in the text are highlighted in yellow.

Reviewer 2

In this paper, the authors fabricated ceramic-metal heterogeneous material by additive manufacturing. Revisions are required before publication:

  1. The effect of ceramics on the microstructure of metals cannot be seen in Fig.3. The author needs to corrode the sample again to characterize the microstructure of titanium alloy, and are further analyze the shape and size of grains.

Response.

The main purpose of the research is to study the mechanical properties of this coating. In our opinion, the general microstructure is of interest, namely, whether the dissolution of the initial ceramic occurs with the formation of secondary phase compounds and how this affects the mechanical properties. From the SEM image and XRF you can see what is happening.

  1. The author should add some critical analysis to improve Introduction. In addition, there are more relevant papers that should be covered in literature review: https://doi.org/10.1016/j.mattod.2022.08.014 https://doi.org/10.1016/j.addma.2021.102462

Response.

Agree with the comment. The literature review takes into account the proposed articles.

  1. There is a problem with the title of Figure 6.

Response.

Agree with the comment. It was corrected.

  1. The author says that ceramics enhance the mechanical properties of titanium alloy. But what about internal defects, such as pores and cracks.

Response.

Agree with the comment. In this work, the laser exposure mode was used, which was found from the results of optimization according to the criterion of the minimum content of pores and cracks. As a result, the formed coating was characterized by the absence of cracks with a small number of pores. A detailed study of the influence of pores on the nature of the destruction of the material will be investigated in future works.

  1. The microhardness of each position was measured only once, and the number of samples was too small.

Response.

In the work, the microhardness values were averaged over three measurements for each distance. Indentation was carried out into the matrix in the area without the initial SiC ceramics. A detailed study of the effect of secondary phase compounds was carried out by the nanoindentation method (a series of 4x100 injections with a step of 20 μm, a total of 400 measurements for each sample).

  1. Please further analyze the mechanism of phase evolution.

Response.

New phases are synthesized in a metal matrix as a result of a chemical reaction between elements (8Ti+3SiC→Ti5Si3+3TiC) during laser exposure. This reaction is possible due to the negative Gibbs free energy, ΔG, which is the maximum energy that is available to do useful work as a result of a chemical reaction.

Reviewer 3 Report

It is the interesting material and not so common. It is something fresh, useful. The Authors presents the results very clearly, SEM images are very good quality. Very impressive are studies of the phase compounds in the bulk of the material carried out using synchrotron radiation (SR) at VEPP-3. I reccommend this paper at this present form. 

Author Response

The authors are grateful to the reviewers for their time and consideration, as well as valuable suggestions for improving the manuscript. All changes in the text are highlighted in yellow.

It is the interesting material and not so common. It is something fresh, useful. The Authors presents the results very clearly, SEM images are very good quality. Very impressive are studies of the phase compounds in the bulk of the material carried out using synchrotron radiation (SR) at VEPP-3. I reccommend this paper at this present form. 

Response.

Thanks to the reviewer for the very high rating of this work.

Reviewer 4 Report

The authors compared the resistance of high-speed impact loads for the heterogeneous materials reinforced by ceramic fiber and powder separately. However, this article focused the attention on the phenomenon description of the materials. For mechanical characteristics, only the hardness and elastic modulus were considered. More references and experiments should be included and analyzed. 

Author Response

The authors are grateful to the reviewers for their time and consideration, as well as valuable suggestions for improving the manuscript. All changes in the text are highlighted in yellow.

Reviewer 4

The authors compared the resistance of high-speed impact loads for the heterogeneous materials reinforced by ceramic fiber and powder separately. However, this article focused the attention on the phenomenon description of the materials. For mechanical characteristics, only the hardness and elastic modulus were considered. More references and experiments should be included and analyzed. 

Response.

Agree with the comment. In the next works, studies on crack resistance and tensile strength will be carried out. After analyzing the existing publications, the team of authors came to the conclusion that at present there are no publications on the resistance of heterogeneous materials created by the AT method, reinforced with ceramic fibers to high-speed impact loads.

Round 2

Reviewer 2 Report

The paper is acceptable.